**Brief Communication**

# Orthology inference at scale with FastOMA

Sina Majidian [1,2], Yannis Nevers [1,2], Ali Yazdizadeh Kharrazi [1], Alex Warwick Vesztrocy [1,2], Stefano Pascarelli [1,2], David Moi [1,2], Natasha Glover[1,2], Adrian M. Altenhoff [2,3] & Christophe Dessimoz [1,2] ✉

The surge in genome data, with ongoing efforts aiming to sequence 1.5 M eukaryotes in a decade, could revolutionize genomics, revealing the origins, evolution and genetic innovations of biological processes. Yet, traditional genomics methods scale poorly with such large datasets. Here, addressing this, 'FastOMA' provides linear scalability for orthology inference, enabling the processing of thousands of eukaryotic genomes within a day. FastOMA maintains the high accuracy and resolution of the well-established Orthologous Matrix (OMA) approach in benchmarks. FastOMA is available via GitHub at https://github.com/DessimozLab/FastOMA/.

Within the decade, the Earth BioGenome initiative aims to sequence 1.5 M eukaryotes[1]. This paves the way for understanding how all species evolved from life's common origin. Yet, due to processing limitations, even the thousands of genomes we have access to today are studied only piecemeal in practice. A fundamental step to comparative genomics analyses is to identify orthologs, genes of common ancestry that originated by speciation events[2]. When performed systematically, orthology delineation conveys how sequences were gained, lost or duplicated, assuming that their basic mode of inheritance is vertical descent. Deriving orthology enables many types of downstream analysis, such as annotation propagation, phylogenomics or phylogenetic profiling[3].

State-of-the-art orthology methods face acute scalability issues[4]. Methods relying on all-against-all sequence comparisons can no longer keep up with today's data, let alone tomorrow's. For state-of-the-art pipelines such as our own Orthologous MAtrix (OMA) algorithm and database[5,6], this translates to >10 million central processing unit (CPU) hours to derive the orthology relationships of >2000 genomes that have been processed thus far. Methods relying on whole-genome alignment, such as TOGA (Tool to infer Orthologs from Genome Alignments)[7], are more efficient, but the genome alignment requirement limits their applicability to relatively closely related species. While small-scale comparative genomics has achieved remarkable progress, a more integrated, large-scale approach would be transformative.

To address this challenge, we introduce FastOMA, which dramatically speeds up orthology inference without sacrificing accuracy or resolution.

FastOMA is a complete rewrite of the OMA algorithm focused on scalability from the ground up (Fig. 1). By combining ultrafast homology clustering using $k$-mers, taxonomy-guided subsampling and a highly efficient parallel computing approach, it achieves linear performance in the number of input genomes. First, we leverage our current knowledge of the sequence universe (with its evolutionary information stored in the OMA database) to efficiently place new sequences into coarse-grained families (hierarchical orthologous groups (HOGs) at the root level) using the alignment-free $k$-mer-based OMAmer tool[8]. In an attempt to detect homology among unplaced sequences (which could belong to families that are absent from our reference database), we then perform a round of clustering using the highly scalable Linclust software[9]. Next, we resolve the nested structure of the HOGs (Supplementary Information 1) corresponding to each ancestor, in an efficient leaf-to-root traversal of the species tree. By avoiding sequence comparisons across different families, the number of computations is drastically reduced compared with conventional approaches (see Methods for details).

FastOMA has high scalability without sacrificing accuracy in a diverse range of benchmarks. We assessed the accuracy of FastOMA on the Quest for Orthologs (QfO) suite of benchmarks[10]. FastOMA retains OMA's high precision accuracy and even improves upon it in terms of recall, positioning it on the Pareto frontier of orthology inference methods. For instance, on the SwissTree reference gene phylogeny benchmark, FastOMA outperforms other methods with a precision of 0.955 in reference gene phylogenies (Fig. 2a). With a recall in line with most state-of-the-art methods (0.69, lower than those of Panther and OrthoFinder), the balance of these metrics indicates a well-tuned approach to orthology inference, with a focus on minimizing false positives. Likewise, on the generalized species tree benchmark at the Eukaryota level, FastOMA is among those with the lowest topological error, with a normalized Robinson–Foulds distance—the number of

[1]Department of Computational Biology, University of Lausanne, Lausanne, Switzerland. [2]Swiss Institute of Bioinformatics, Lausanne, Switzerland. [3]Department of Computer Science, ETH Zurich, Zurich, Switzerland. ✉e-mail: Christophe.Dessimoz@unil.ch

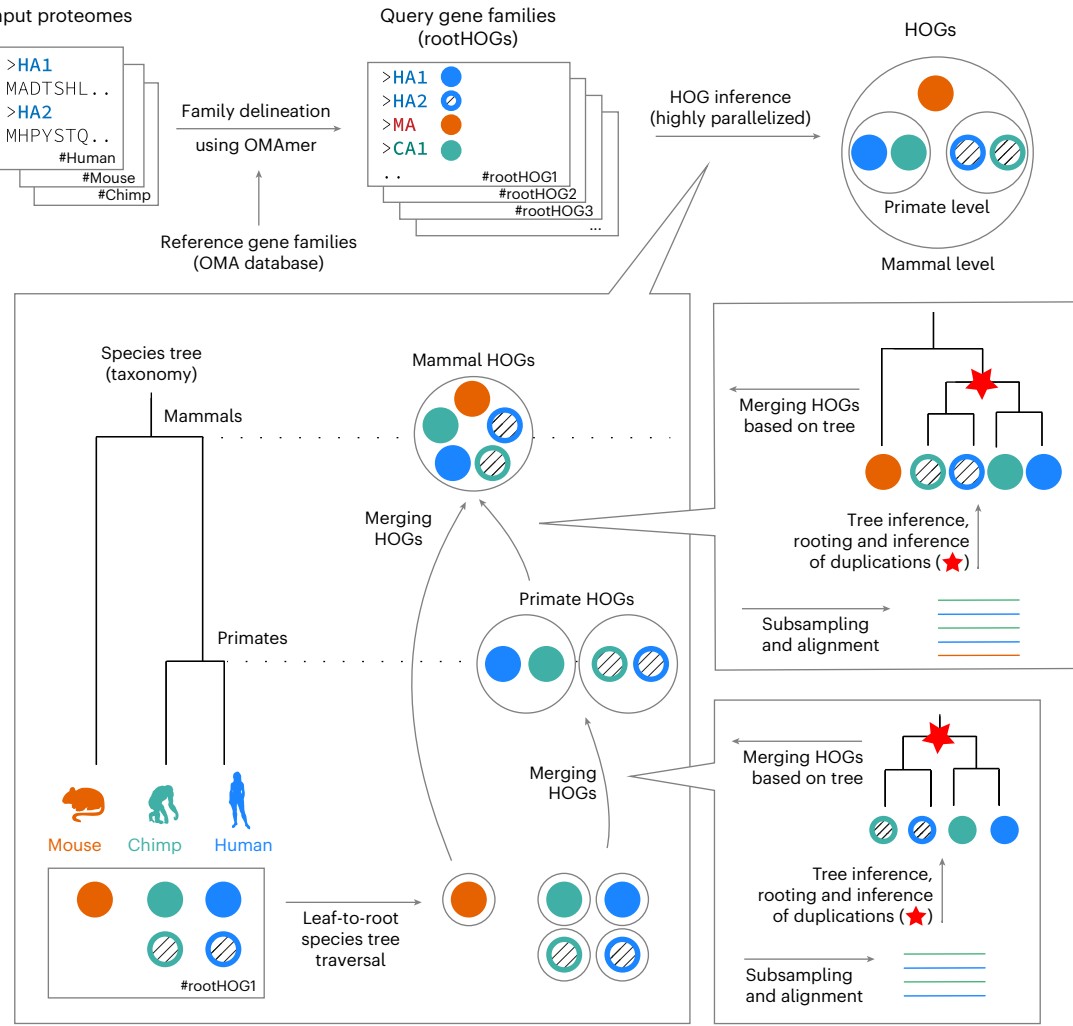

**Fig. 1 | FastOMA algorithm overview.** Input proteomes are mapped to reference gene families using the OMAmer software, forming hierarchical orthologous groups (HOGs) at the root level (rootHOGs), see Methods. HOGs are inferred using a 'bottom-up' approach, starting from the leaves of the species tree and moving towards the root. At each taxonomic level, HOGs from the child level are merged, resulting in HOGs at the current level. To decide which HOGs should be merged, sequences from the child HOGs are used to create a MSA[18], followed by gene tree inference[19] to identify speciation and duplication events[20]. Child HOGs are merged if their genes evolved through speciation (see Methods and Supplementary Information 1 for details). Credit: human silhouette, T. Michael Keesey (Public Domain Mark 1.0); chimpanzee silhouette, Jonathan Lawley (CC0 1.0 Universal); mouse silhouette, Soledad Miranda-Rottman (CC BY 3.0), PhyloPic.

different edges between two trees normalized by the total number of internal edges—of 0.225 to the reference tree, at moderate recall (Fig. 2b and Supplementary Information 2–17).

A key achievement of FastOMA is its linear scaling behavior (Fig. 2c), which opens up the possibility of processing extensive datasets rapidly. FastOMA inferred orthology among all 2,086 eukaryotic UniProt reference proteomes in under 24 h, using 300 CPU cores. In the same timespan, the original OMA algorithm could process only 50 genomes. Even methods optimized for speed such as OrthoFinder[11] or SonicParanoid[12] still exhibit quadratic time complexity (Fig. 2c). Thus, FastOMA's linear scalability breaks new ground.

The initial sequence placement step using OMAmer helps FastOMA achieve its speed, but the subsequent alignment and tree inference steps are critical for its accuracy. Indeed, sequence placement alone is considerably less accurate than state-of-the-art methods in benchmarks (Supplementary Information 3).

FastOMA exploits known taxonomic relationships to reduce the number of sequence comparisons. By default, it relies on the commonly used National Center for Biotechnology Information (NCBI) taxonomy[13,14], but users can specify any reference species phylogeny as input. To assess the impact of the resolution of the input tree on orthology accuracy, we compared FastOMA's performance on UniProt reference proteomes with a more resolved species tree derived from the TimeTree resource[15]. Compared with the NCBI taxonomy, this resulted in improved ortholog predictions, with more parsimonious gene family evolution history, lowering the number of implied gene losses across all gene families (Fig. 2d). FastOMA is also robust to errors artificially introduced in the species taxonomy (Supplementary Figs. 18–20). FastOMA can thus use advances in taxonomic knowledge for better orthology predictions and will benefit from the higher resolution that is brought by new genomic sequences from large-scale sequencing projects.

FastOMA contains additional features that make it easier to deal with complex and noisy genomic data. It is designed to handle multiple isoforms for the genes resulting from alternative splicing and select the most evolutionarily conserved ones, and can also deal with fragmented gene models[16]. Both features lead to noticeable improvements in FastOMA inferences (Supplementary Information 9 and 10). As it uses the

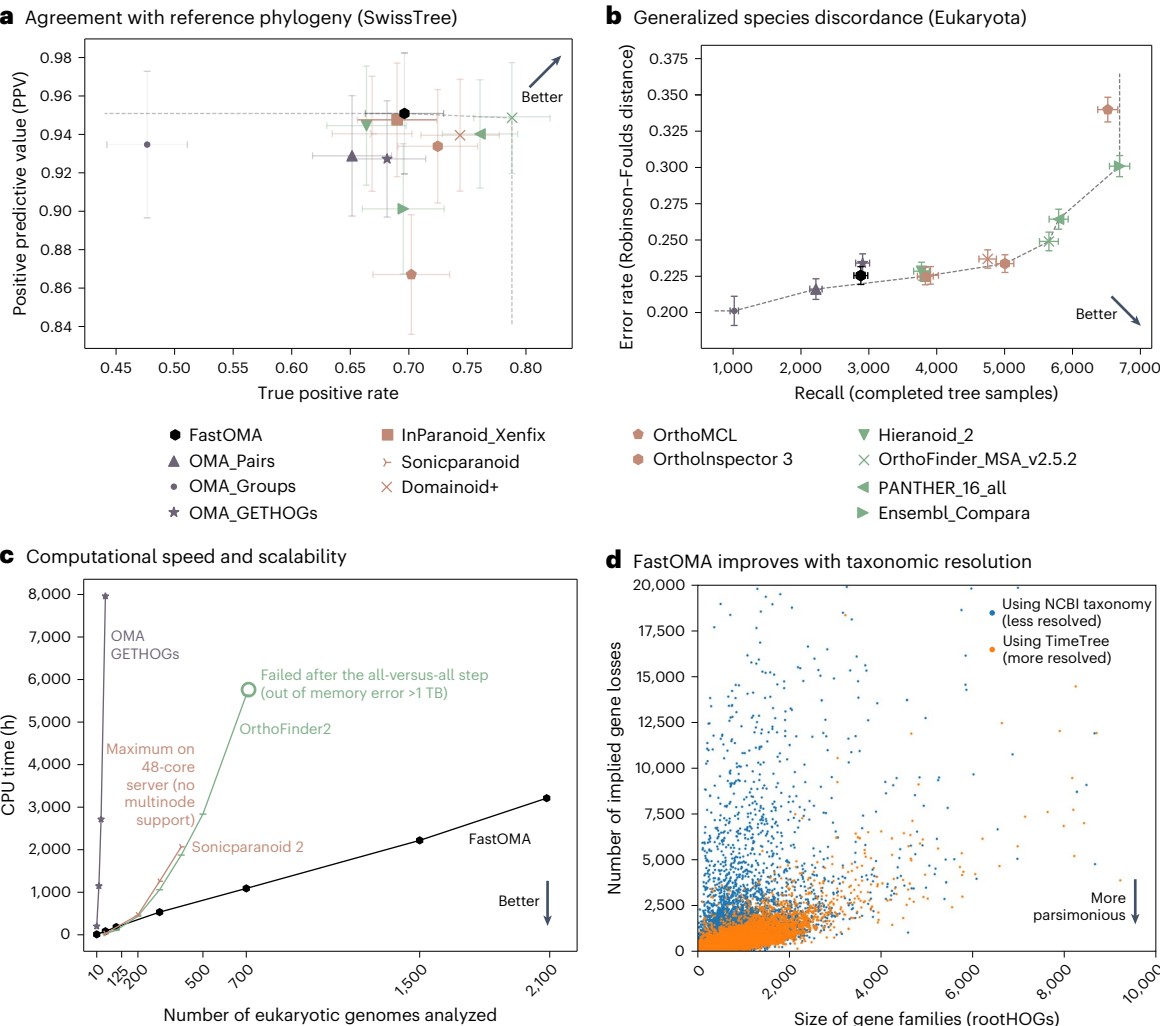

**Fig. 2 | FastOMA is not only fast but also accurate. a,** QfO benchmark[10], agreement with SwissTree reference phylogeny covering 19 manually curated gene trees. The error bars indicate 95% confidence intervals comparing FastOMA with EnsemblCompara[21], Domainoid[22], OrthoMCL[23], OrthoInspector[24], sonicparanoid, PANTHER[25], OrthoFinder, Hieranoid[26] and the OMA family including OMA pairs, OMA groups and OMA GETHOGs (graph-based efficient technique for HOGs)[27–29]. **b,** QfO benchmarking of the generalized species discordance test on the Eukaryota clade, where the gene tree inferred from orthologous genes is compared with the reference species tree considering up to 3,000 gene trees per method (see Supplementary Information 2.1 for details). **c,** A computation time comparison of FastOMA and state-of-the-art alternatives. **d,** The impact of species tree resolution on the complexity of the gene family evolutionary scenario (proxied by the number of gene losses over the gene family history). Each point represents a gene family (a rootHOG), whereby the size of a gene family corresponds to the number of genes in it[30] (the figure is truncated to focus on the most relevant region; see Supplementary Fig. 24 for a version with all data, and see Methods for the implied losses calculation).

same data structure as OMA, FastOMA benefits from its rich ecosystem of downstream applications, including phylogenetic profiling, efficient gene family visualization, ancestral synteny inference and advanced phylostratigraphy, enabling researchers to trace gene family histories and understand gene emergence, duplication and loss events[5,17].

In conclusion, the FastOMA algorithm offers a unique solution for accurate orthology inference, making it possible to study evolutionary history at the scale of massive genomics projects. Future work will aim to further refine orthology inference by integrating structural protein data to improve resolution at deeper evolutionary levels, as well as gene order conservation as an additional layer of information.

## Online content

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

## Methods

### FastOMA algorithm outline

FastOMA is a method for inferring orthology relationships. The input to FastOMA includes the proteome sets of species and the species tree. The FastOMA algorithm consists of two main steps: finding rootHOGs and inferring the nested structure of HOGs (Fig. 1).

### Step 1: FastOMA gene family inference

The FastOMA algorithm infers gene families from the provided proteomes. The process begins by mapping the input proteomes onto the reference HOGs (Supplementary Information 1) using the OMAmer tool (Fig. 1). Proteins mapped to the same reference HOG are then grouped together, forming query rootHOGs, with the exclusion of proteins already present in the database. Thus, proteins in the database reference HOGs are not used in the next steps in FastOMA.

Although each rootHOG ideally represents a single gene family, instances may arise where a gene family of query proteomes is split among multiple rootHOGs. To address this, FastOMA tries to find those query rootHOGs that are associated with the same gene family. FastOMA leverages the ability of OMAmer to report multiple rootHOGs to which the sequences could be mapped, along with their score. This score ('family_p') is the P value of having as many or more $k$-mers in common between the protein sequence and the HOG under a binomial distribution, reported in negative natural logarithm. Considering a minimum threshold of 70 (by default), we construct a graph of rootHOGs, where each node represents a query rootHOG. In such a graph, we add an edge between two nodes (rootHOGs) when a minimum of ten proteins (by default) are mapped to both query rootHOGs and it represents at least either 80% of all proteins mapping to the bigger rootHOG or 90% of those mapping to the smallest one. This ensures a high overlap of protein content of the merged rootHOG. Finally, we group the members of all HOGs in each highly connected component of this graph in a single query rootHOG.

It is worth noting that some proteins may not be assigned to any reference HOGs owing to no recognizable homologs in the reference database. In addition, there is a scenario where only one protein is mapped to the rootHOG, referred to as a singleton, representing an individual rather than a group[5]. To ensure those genes are not lost to FastOMA's orthology inference, these singletons and unmapped sequences are combined into a FASTA file on which we run Linclust, the clustering tool from the MMseqs package[9]. This yields new query rootHOGs.

Critically, assigning proteins to rootHOGs (gene families) allows us to avoid unnecessary all-against-all comparisons of unrelated proteins (those without homology), thanks to the speed of OMAmer and Linclust. All the query rootHOGs are written as FASTA files to be used in the next step and can be handled in parallel.

Notably, the OMA team provides regular updates to the OMA database, increasing the number and diversity of species included in the database used by OMAmer. This results in higher resolution for $k$-mer-based grouping. As more taxa get included, we foresee FastOMA's inference will improve as more sequences are placed into rootHOGs.

### Step 2: FastOMA orthology inference

For every query rootHOG, FastOMA infers the nested structure of the HOG (as depicted in Fig. 1). The objective is to identify the genes that are grouped together at each taxonomic level as a HOG, which means they descended from a single gene at that specific level. Note that the number of HOGs at each level reflects the number of copies of the gene present in the ancestral species.

To achieve this, FastOMA follows a bottom-up approach by traversing the species tree. Starting from the leaves of the tree (extant species), each gene in the species' proteome is treated as a HOG. At each level in the traversal, certain HOGs from the child level are combined. The determination of which HOGs will be merged is guided by a gene tree containing the proteins of species descending from this node. The merging is done for all HOGs that descended from the same common ancestor by a speciation event. The entire process is detailed below:

**Gene tree inference.** All the proteins in HOGs at the child level are collectively used for generating a multiple sequence alignment (MSA) using the MAFFT package[18]. As part of the FastOMA Python script, the MSA undergoes column-wise trimming with a default threshold of 0.2, meaning that we remove columns of the MSA that have more than 80% gap elements (Supplementary Information 5). Aligned sequences (rows in MSA) that exceed a default threshold of >50% gaps are subsequently removed. However, we keep them in the HOG, but they are not used for tree inference. Subsequently, we employ FastTree[19] to infer the gene tree, and this tree is rooted using the midpoint approach.

To expedite the orthology inference process at deeper levels of the trees where the number of children is prohibitively high, we implement a subsampling approach, retaining only a specified number of proteins per HOG (Supplementary Figs. 12–14; by default, 20 proteins are randomly selected) used for the MSA and tree inference. The unsampled sequences will have the same fate as the rest of the proteins in the same group at the defined taxonomic level.

Note that the subsampling strategy is key to the speed of FastOMA, and expectedly, there is a trade-off between accuracy and speed. Our benchmarking results indicate that FastOMA performs well with the subsampling approach, but users can change the degree of the subsampling in the parameter file.

**Duplication and speciation event labeling.** Each internal node in the gene tree is classified as either a duplication or a speciation event using the species overlap method[20]. For each node in the gene tree, this involves calculating the ratio of the number of shared species between its two subtrees divided by the number of all species (union). If the ratio equals zero, the node is labeled as a speciation event; otherwise, it is labeled as a duplication event. When the species overlap ratio is less than 0.1 (as per default settings), indicating very low support for a duplication event, all leaves from the child subtree with the least number of proteins are excluded from merging decisions (described in 'HOG merging' section). In other words, these proteins will stay in the corresponding HOGs as in the previous taxonomic level, and only the taxonomic label of the HOG is updated to the current taxonomic level (assuming no other merging happens in another part of the gene tree for this HOG). This is done to ensure that errors in gene annotation or inaccurate tree inference only minimally affect the orthology inference.

**HOG merging.** Starting from the root of the gene tree, evidence of a speciation event (that is, the internal node annotated as a speciation event due to no species overlap) prompts the merging of the HOGs of the leaves descending from the nodes. This is achieved by constructing a HOG graph, where each node represents a HOG. An edge is introduced between HOG1 and HOG2 if protein 1 (located in HOG1) and protein 2 (in HOG2) coalesce at a speciation event in the gene tree. Subsequently, each connected component within this graph constitutes a HOG at the current level of the species tree. Furthermore, FastOMA has a mechanism to handle spuriously merged subHOGs; at the deeper taxonomy level, when genes within a subHOG coalesce at a duplication event in the gene tree, FastOMA splits the subHOG into two, ensuring copies of ancestral genes are not co-present in a subHOG.

**Inferring orthology relationship.** Once the species tree traversal is complete, the nested structure of the query HOG is fully resolved. From the HOG structure inferred this way, all orthology and paralogy relationships can be efficiently deduced.

**Note on parallelization.** Scalability has been a major challenge in the field of orthology inference highlighted by the QfO community for many years[4,10,31]. FastOMA is optimized to process taxonomic levels in parallel (when possible) by inferring HOGs at all taxonomic levels, accounting for dependencies among child HOGs, that is, a node will be processed after all its child nodes are processed. To optimize parallelization efficiency by avoiding unnecessary overheads of Nextflow and Slurm management workflows, FastOMA groups approximately 150 small- to medium-sized query rootHOGs together, treating them as a single job. Conversely, large rootHOGs are processed individually (to infer nested structure of HOGs) for optimal performance using Python-future for which taxonomic parallelization is activated. The default rootHOG file size threshold for this purpose is 400,000 bytes, or ~500 proteins (Supplementary Information 11).

## FastOMA outputs
The main output of FastOMA is an OrthoXML file that stores HOGs and their nested structures, allowing to reconstruct their evolutionary histories. Furthermore, FastOMA reports the protein list in each rootHOG (gene family) in TSV format. A final FastOMA output is a list of proteins in strict orthologous groups, wherein all genes within the group are orthologous to each other, which can be used as marker genes for phylogenetic analyses[32,33]. Besides, the user can store the gene trees and MSAs of the subsampled HOGs for all taxonomic levels.

## Isoform selection
FastOMA is capable of handling proteomes that feature multiple protein isoforms for a gene due to alternative splicing. Users can provide an isoform file where each row lists comma-separated protein IDs associated with a gene. FastOMA selects the isoform with the highest 'family_p' score, the one with the best fit to known proteins in the reference rootHOG based on $k$-mer content. For the evaluation of isoform selection, we used the UniProt reference proteomes and their splice information (https://ftp.uniprot.org/pub/databases/uniprot/current_release/knowledgebase/reference_proteomes/Eukaryota).

## FastOMA software
The FastOMA codebase is composed of multiple subpackages written in Python. FastOMA benefits from the Nextflow workflow to parallelize different steps and subpackages considering the dependencies modeled as a direct acyclic graph (Supplementary Information 11). The software is publicly available on GitHub (https://github.com/DessimozLab/FastOMA) and on DockerHub (https://hub.docker.com/r/dessimozlab/fastoma).

## Time comparison on eukaryotic dataset
We considered all 2,181 eukaryotic UniProt reference proteomes (accessed on 25 January 2023) and filtered them to keep those with a minimum BUSCO (benchmarking universal single-copy orthologs) completeness of 50%, resulting in 2,086 proteomes in total. We ran SonicParanoid, OrthoFinder and FastOMA on datasets with different sizes ranging from 10 to 2,086 species. OrthoFinder 2.5.4 was run in two steps. First, to generate all-against-all sequence comparisons, we used the -op parameter to generate and execute command lines for Diamond. Then, the rest of OrthoFinder was conducted. SonicParanoid 2.0.4 was used with default parameters using 48 CPUs with a limit of 3 days wall clock. It is neither possible to parallelize SonicParanoid2 on different computation nodes nor to feed it with the result of Diamond; hence, we could not obtain compute time for the larger datasets during the mentioned time limit. For FastOMA, the NCBI tree was used by downloading via the ETE3 package[14]. The comparison of tools in terms of wall-clock time in hours is reported in Supplementary Fig. 25. The Diamond part of OrthoFinder and all steps of FastOMA use different nodes on the cluster, so the reported wall-clock time might have been affected by the availability of nodes at the time of each run. However, the CPU times reported in Fig. 2c are more accurate.

## Analysis on tree resolution
We ran FastOMA on both the TimeTree and the NCBI tree. For the Time-Tree analysis, we uploaded the list of species names to the TimeTree webserver[15] (https://timetree.org). This resulted in a species tree with 1,757 leaves since some of the species were not available in TimeTree. We ran FastOMA with default parameters on the dataset of 1,757 proteomes and with both the TimeTree tree and NCBI tree as the species tree. We used pyHAM[30] for calculating the implied gene losses.

To calculate the estimated proportion of proteomes composed of fragments, we ran OMArk[16] v0.3 on all proteomes. We used the BUSCO statistics downloaded from the UniProt website for the full eukaryotic dataset.

We also conducted another analysis to study the impact of the species tree for the QfO dataset where five pairs of species are swapped. The results are provided in Supplementary Information 7 and Supplementary Figs. 18–20, where FastOMA shows a moderate level of robustness. However, having an erroneous species tree impacted the orthology inference by introducing false positives.

To conclude, we highlight that the orthologous and paralogous genes are found using the species overlap method on the gene tree and the species tree is used to determine the order of comparisons, defining the HOG structure. Thus, a fully resolved species tree is not needed to infer orthology information with FastOMA. However, errors in the species tree can potentially propagate through the orthology inference process.

## Benchmarking against the QfO reference proteome set
We ran FastOMA on the 78 reference proteomes used in the QfO benchmark and the associated standard species trees as input. We then submitted the results to the QfO benchmarking service[4,10,31,34] and obtained the results on the 11 available benchmarks. In these benchmarks, FastOMA is compared with several state-of-the-art methods that are available in the QfO public resource, including EnsemblCompara[21], Domainoid[22], OrthoMCL[23], OrthoInspector[24], sonicparanoid[35], PANTHER[25], OrthoFinder[11], Hieranoid[26] and the OMA family[27–29,36]. QfO analysis is described in detail in Supplementary Information 2. The orthogroup benchmarking for the clade Bilateria[37] is provided in Supplementary Information 8.

## Analysis of the QfO reference proteome set using InterProScan classification of protein families
To study the influence of the OMA database and OMAmer on the performance of FastOMA, we replaced the first part of the procedure, normally done by placing query genes into the OMA database rootHOGs with OMAmer, with InterProScan. We used InterProScan to group the QfO proteomes into gene families predefined by InterProScan[38]. To do so, we first ran InterProScan with the argument -appl Pfam on the QfO dataset, which grouped the proteins into InterProScan families[39]. Then, we created the rootHOG with those groups, maintaining the same InterProScan family identifier. Then, we ran the rest of FastOMA on these rootHOG FASTA files. The QfO benchmarking results are shown in Supplementary Information 6 and Supplementary Figs. 15–17. Note that a user can provide their own initial grouping of proteins to be used with FastOMA. This could be put in practice in two ways: (1) running the last two processes of FastOMA.nf (hog_rest and collect_subhog) on the user's protein family in FASTA format or (2) providing group mapping of proteins in the OMAmer format.

## Computations
All the analyses were conducted on the high-performance computer cluster of the University of Lausanne that houses 96 computation nodes. Each node is equipped with two 24-core AMD (Advanced Micro Devices) CPUs, totaling 48 cores per node. Data were written and read on a 150 TB SSD (solid-state drive) scratch drive. For the QfO analysis, most steps of FastOMA needed less than 10 GB of memory, with a maximum of 32 GB.

**Reporting summary**

Further information on research design is available in the Nature Portfolio Reporting Summary linked to this article.

## Data availability

UniProt reference proteomes and splice information (_additional.fasta.gz) were downloaded from https://ftp.uniprot.org/pub/databases/uniprot/current_release/knowledgebase/reference_proteomes/Eukaryota. The 2020 version of QfO proteomes was downloaded from the EBI repository at http://ftp.ebi.ac.uk/pub/databases/reference_proteomes/previous_releases/qfo_release-2020_04_with_updated_UP000008143/. The OMAmer database used in this study is available at https://oma-browser.org/All/LUCA.h5. The OMAmer database, an archive of FastOMA code, the TimeTree with annotation of internal nodes of 1,757 species in Newick format, the UniProt IDs and the inferred HOG for 1,757 eukaryotic species in OrthoXML format are all deposited on Zenodo at https://doi.org/10.5281/zenodo.10403053 (ref. 40).

## Code availability

FastOMA is free open-source software (Mozilla Public License 2.0) available via GitHub at https://github.com/DessimozLab/FastOMA. We used the publicly available code for the QfO benchmarking test which is available via GitHub at https://github.com/qfo/benchmark-webservice. A copy of the FastOMA software is available via Zenodo at https://doi.org/10.5281/zenodo.10403053 (ref. 40).

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

## Acknowledgements

We thank C. Train for updating PyHam, as well as B. Sipos and S. K. Bhurji for helpful feedback on FastOMA. This work was funded by the Swiss National Science Foundation (grant 205085) to C.D.

## Author contributions

S.M., A.M.A. and C.D. developed the method. S.M. and A.M.A. implemented the software. S.M., A.Y.K., Y.N., A.W.V., N.G., D.M. and S.P. contributed to the analysis. C.D. and S.M. wrote and edited the manuscript. All authors read and approved the final version of the manuscript.

## Competing interests

The authors declare no competing interests.

## Additional information

**Correspondence and requests for materials** should be addressed to Christophe Dessimoz.

# Reporting Summary

## Statistics

For all statistical analyses, confirm that the following items are present in the figure legend, table legend, main text, or Methods section.

| n/a | Confirmed | |
|---|---|---|
| ☐ | ☒ | The exact sample size (*n*) for each experimental group/condition, given as a discrete number and unit of measurement |
| ☐ | ☒ | A statement on whether measurements were taken from distinct samples or whether the same sample was measured repeatedly |
| ☐ | ☒ | The statistical test(s) used AND whether they are one- or two-sided<br>*Only common tests should be described solely by name; describe more complex techniques in the Methods section.* |
| ☒ | ☐ | A description of all covariates tested |
| ☒ | ☐ | A description of any assumptions or corrections, such as tests of normality and adjustment for multiple comparisons |
| ☐ | ☒ | A full description of the statistical parameters including central tendency (e.g. means) or other basic estimates (e.g. regression coefficient) AND variation (e.g. standard deviation) or associated estimates of uncertainty (e.g. confidence intervals) |
| ☐ | ☒ | For null hypothesis testing, the test statistic (e.g. *F*, *t*, *r*) with confidence intervals, effect sizes, degrees of freedom and *P* value noted<br>*Give P values as exact values whenever suitable.* |
| ☒ | ☐ | For Bayesian analysis, information on the choice of priors and Markov chain Monte Carlo settings |
| ☒ | ☐ | For hierarchical and complex designs, identification of the appropriate level for tests and full reporting of outcomes |
| ☒ | ☐ | Estimates of effect sizes (e.g. Cohen's *d*, Pearson's *r*), indicating how they were calculated |

*Our web collection on statistics for biologists contains articles on many of the points above.*

## Software and code

Policy information about availability of computer code

| Data collection | ETE3 v3.1.3 for downloading NCBI tree. |
|---|---|
| Data analysis | SonicParanoid v2.0.4, OrthoFinder 2.5.4, OMArk v0.3,  pyHAM v1.2.0, InterProScan 5.52, FastOMA v0.3.2, omamer v2.0.4, Seaborn v0.13.2, scipy v1.14.0. |

For manuscripts utilizing custom algorithms or software that are central to the research but not yet described in published literature, software must be made available to editors and reviewers. We strongly encourage code deposition in a community repository (e.g. GitHub). See the Nature Portfolio guidelines for submitting code & software for further information.

## Data

Policy information about availability of data

All manuscripts must include a data availability statement. This statement should provide the following information, where applicable:
- Accession codes, unique identifiers, or web links for publicly available datasets
- A description of any restrictions on data availability
- For clinical datasets or third party data, please ensure that the statement adheres to our policy

UniProt reference proteomes and splice information (_additional.fasta.gz) were downloaded from https://ftp.uniprot.org/pub/databases/uniprot/current_release/knowledgebase/reference_proteomes/Eukaryota. The 2020 version of QfO proteomes was downloaded from the EBI repository at http://ftp.ebi.ac.uk/pub/databases/reference_proteomes/previous_releases/qfo_release-2020_04_with_updated_UP000008143/. The OMAmer database used in this study is available at

## Human research participants

Policy information about studies involving human research participants and Sex and Gender in Research.

| | |
|---|---|
| Reporting on sex and gender | N/A |
| Population characteristics | N/A |
| Recruitment | N/A |
| Ethics oversight | N/A |

Note that full information on the approval of the study protocol must also be provided in the manuscript.

## Field-specific reporting

Please select the one below that is the best fit for your research. If you are not sure, read the appropriate sections before making your selection.

☐ Life sciences　　☐ Behavioural & social sciences　　☒ Ecological, evolutionary & environmental sciences

For a reference copy of the document with all sections, see nature.com/documents/nr-reporting-summary-flat.pdf

## Ecological, evolutionary & environmental sciences study design

All studies must disclose on these points even when the disclosure is negative.

| | |
|---|---|
| Study description | We developed a tool for comparing the proteomes of different species. |
| Research sample | Eukaryotic reference proteomes available on UniProt website. |
| Sampling strategy | N/A |
| Data collection | We analyzed all the Eukaryotic reference proteomes available on UniProt website. |
| Timing and spatial scale | N/A |
| Data exclusions | N/A |
| Reproducibility | *Describe the measures taken to verify the reproducibility of experimental findings. For each experiment, note whether any attempts to repeat the experiment failed OR state that all attempts to repeat the experiment were successful.* |
| Randomization | N/A |
| Blinding | N/A |

Did the study involve field work?　☐ Yes　☒ No

## Reporting for specific materials, systems and methods

We require information from authors about some types of materials, experimental systems and methods used in many studies. Here, indicate whether each material, system or method listed is relevant to your study. If you are not sure if a list item applies to your research, read the appropriate section before selecting a response.

## Materials & experimental systems

| n/a | Involved in the study |
|-----|----------------------|
| ☒ | ☐ Antibodies |
| ☒ | ☐ Eukaryotic cell lines |
| ☒ | ☐ Palaeontology and archaeology |
| ☒ | ☐ Animals and other organisms |
| ☒ | ☐ Clinical data |
| ☒ | ☐ Dual use research of concern |

## Methods

| n/a | Involved in the study |
|-----|----------------------|
| ☒ | ☐ ChIP-seq |
| ☒ | ☐ Flow cytometry |
| ☒ | ☐ MRI-based neuroimaging |

