## [Peer Review File · Nature Methods]

Orthology inference at scale with FastOMA

Corresponding Author: Professor Christophe Dessimoz

Version 0:

Decision Letter:

20th Mar 2024

Dear Professor Dessimoz,

Your Brief Communication, "Orthology inference at scale with FastOMA", has now been seen by 3 reviewers. As you will see from their comments below, although the reviewers find your work of potential interest, they have raised a number of concerns. We are interested in the possibility of publishing your paper in Nature Methods, but would like to consider your response to these concerns before we reach a final decision on publication.

We therefore invite you to revise your manuscript to fully address all these concerns. In light of Reviewer 3's comments, we strongly encourage you to demonstrate other advantages in FastOMA compared with other tools besides speed.

Link Redacted

We hope to receive your revised paper within 3 months. If you cannot send it within this time, please let us know. In this event, we will still be happy to reconsider your paper at a later date so long as nothing similar has been accepted for publication at Nature Methods or published elsewhere.

OPEN SCIENCE REQUIREMENTS

REPORTING SUMMARY AND EDITORIAL POLICY CHECKLISTS

DATA AVAILABILITY

All novel DNA and RNA sequencing data, protein sequences, genetic polymorphisms, linked genotype and phenotype data, gene expression data, macromolecular structures, and proteomics data must be deposited in a publicly accessible database, and accession codes and associated hyperlinks must be provided in the "Data Availability" section.

CODE AVAILABILITY

Please include a "Code Availability" subsection in the Online Methods which details how your custom code is made available. Only in rare cases (where code is not central to the main conclusions of the paper) is the statement "available upon request" allowed (and reasons should be specified).

MATERIALS AVAILABILITY

ORCID

Nature Methods is committed to improving transparency in authorship. As part of our efforts in this direction, we are now

requesting that all authors identified as 'corresponding author' on published papers create and link their Open Researcher and Contributor Identifier (ORCID) with their account on the Manuscript Tracking System (MTS), prior to acceptance. This applies to primary research papers only. ORCID helps the scientific community achieve unambiguous attribution of all scholarly contributions. You can create and link your ORCID from the home page of the MTS by clicking on 'Modify my Springer Nature account'. For more information please visit www.springernature.com/orcid.

Sincerely,

Lin Tang, PhD
Senior Editor
Nature Methods

Reviewers' Comments:

Reviewer #1:

Remarks to the Author:

The manuscript presents FastOMA, a new method building on the OMA framework, to infer orthologous genes across large datasets.

The most important feature is the linear scalability with respect to the number of input species, which makes FastOMA a method that can handle thousands of genomes. At the heart of this speed improvement is replacing the all-vs-all alignment step with a k-mer based grouping of new genes into existing homology groups (OMAmer) and efficient parallelization over the ortholog groups and the phylogenetic tree. The authors show that FastOMA achieves a high accuracy, comparable to the best methods available. The method can also handle isoforms by picking the one that best matches a protein in the reference HOG, and benefits from more accurate phylogenies.

The method is an asset in an era where hundreds of genomes will (hopefully) soon be generated per day. The paper is well written and well explained. I only have minor comments and suggestions.

- My major question is how the method works in case the input tree has a polytomy (not sure if a strictly bifurcating tree is expected; the NCBI taxonomy (line 85) is certainly not) or has an error. E.g. if the given tree includes ((A,B)C) as a clade, but in reality it is (A(B,C)), this will likely cause an error in the gene tree reconciliation step for A+B. Would this get fixed when A+B+C is considered as a clade, or would the early error be propagated up the tree.

Maybe robustness to tree error could also be tested by introducing errors in the input tree, and determining how accuracy drops.

- OMAmer requires reference HOGs as input and the manuscript in line 41 (current knowledge of the sequence universe) also mentions this. It would be great to make clear in Figure 1, which input are new input proteomes and which are reference HOGs (gene families).

- line 63. It would be more informative to report the number of cores (not CPUs), as CPUs nowadays have many cores. In line 297 the authors likely mean 96 nodes each equipped with an AMD CPU with 48 cores.

- Line 28: Scalability issues and all-vs-all. While this is true for almost all methods, there is one orthology inference method available that scales linear with the number of species (<https://www.science.org/doi/full/10.1126/science.abn3107>)

- Line 626 mentions that fragmented gene models can arise due to genome annotation errors. Another likely source is assembly fragmentation, where the gene is split across 2 or more scaffolds. While this rarely a problem for long-read based assemblies, most of the existing assemblies are based on short reads and this error is quite prevalent.

- As a note, methods like FastOMA will benefit from more comprehensive genomes covering different taxonomic groups, as this will make k-mer approaches more efficient (simply by reducing the sequence divergence between species pairs) and reducing the number of proteins not assigned to a reference HOG. This forward-looking point could be mentioned. Along a similar line, phylogenetic resolution and thus input tree knowledge will improve when more species have been sequenced.

Remarks on code availability:

While I didn't test FastOMA myself, the github is very well documented and provides test data. The code looks reasonably well documented and clean.

Various options are available for running it via docker, singularity or on clusters via Slurm.

Reviewer #2:

Remarks to the Author:

The manuscript describes the FastOMA method for orthology inference. Input is a set of proteomes, each comprising the protein sequences encoded in the genome for a certain species, and a species tree representing the evolutionary relationships between the input species. The output is a hierarchy of clusters of orthologous proteins, with each hierarchy level corresponding to an ancestral node in the species tree.

Crucially, FastOMA is the first software tool capable of inferring orthology groups for thousands of proteomes, thanks to the fact that its computation time scales only linearly, instead of quadratically as the state-of-the-art tools, with the number of input proteomes (Fig. 2c). The key idea is to make use an existing database of orthologous groups to which each sequence is mapped, avoiding all-versus-all searches. Only sequences that cannot assigned to an existing orthology group are clustered to found new clusters of orthologs.

FastOMA's unequalled speed and scaling will be critical to analyze the hundreds of thousands of eukaryotic species that are going to be sequenced over the next years, for instance within the Earth Biogenome project.

The tool is very well described in the online methods and supplemental methods sections.

The recall and precision of the inferred orthology relationships are benchmarked using the Quest for Orthologs suite of benchmark by running the publicly available benchmarking scripts with the 2020 version of QFO proteomes. While this code was developed by the same lab of Dessimoz (Altenhoff, A. M. et al. Standardized benchmarking in the quest for orthologs. *Nat. Methods* 13, 425–430 (2016)), the benchmarks have been developed jointly by the Quest for Orthologs community and approved by them. Therefore, I trust that the benchmarks afford a nuanced and balanced assessment of the various orthology inference tools.

The performance of FastOMA in the various QFO benchmarks is shown in the main text and many plots in the supplemental material. Generally speaking, FastOMA is comparable in performance to OMA and is at or near the Pareto frontier of a large number of tools benchmarked by QFO.

In summary, FastOMA is the first tool that scales linearly in the number of input proteomes. It is the only tools that can process thousands and probably even tens of thousands of proteomes while maintaining a competitive sensitivity-recall performance. Its speed leaves no alternatives when huge datasets of proteomes are to be analyzed at once.

Minor points:

1. How much does FastOMA's performance depend on the existing OMA database? It would be insightful to run FastOMA without this database on a few benchmark datasets to assess the contribution of the OMA database to its accuracy and how much it depends on the overlap of the input proteome set with the proteomes in the OMA database.
2. As FastOMA's performance will depend to some extent on the size and completeness of OMA, it would be helpful to discuss the procedure and frequency of updates of the OMA database.
3. Misspelled Schilcker => Schlicker; evolutionar*il*y conserved; Fig. 1a and 1b referred to but a and b labels missing in figure 1.

Reviewer #3:

Remarks to the Author:

This manuscript presents FastOMA, a reimplement of OMA with a focus on speed while maintaining the accuracy of OMA. This is obtained by a new algorithm to reduce the time complexity from quadratic to linear in terms of the number of sequences. However, I have several concerns/comments below.

1. Given that OMA is not as popular as other methods like OrthoFinder, I'm wondering what would be the impact of FastOMA? For example, apart from speed, what other advantage(s) do you have in FastOMA compared with OrthoFinder and BUSCO? While the speedup is impressive, I believe the authors should provide other arguments to justify its publication in *Nature Methods*.
2. Figure 1 is hard to follow and the current caption doesn't really help. The authors should spend more effort to explain it, either in caption or main text or both.
3. Figure 2 is very complicated. For example, 2a is very busy because authors try to condense two accuracy statistics into one single figure. I would suggest to either remove several methods (e.g., keeping only those in Fig 2c) or changing it to just show one statistic on the y-axis and different methods on the x-axis (such as a barplot) or both, which will show the comparison between methods much clearer. Whereas you can move the plot for all methods into the supplement.
4. Fig 2a shows that FastOMA is not as accurate as OrthoFinder (at least in terms of true positive rate – the x-axis). I believe this should be stated in the ms.
5. I don't understand fig 2b and 2d.

6. Page 2, Line 76: Authors should explain what is “normalised Robinson Foulds distance” for non-phylogeneticists.
7. y-axis label of 2b is misleading: Precision is not Robinson Foulds distance, because readers would expect higher precision to mean better, which is not the case here.
8. Page 2, line 63: authors say it took under 24 hours using 300 CPUs, which is not clear how this is possible, because on page 8 line 296-299, they are using 96 nodes each with 48 CPUs. Is that because the authors can distribute the computation across different nodes? And how many CPU cores per node are used? Please clarify.
9. Figure 2c: I would show the wall-clock times instead of CPU times, because it gives readers an impression that it takes 3000 hours (125 days) to get the results, but in fact only 24 hours to wait for the result as written in the main text.
10. Subfigures in Figure 2 should be reordered by reference in the main text. For example, 2c should become 2a as 2c is mentioned first in the text.
11. Page 13, From the supplementary, the experiments were performed in considerable large scale, with over 984K protein among 78 species, including Eukaryota, Fungi, Bacteria, etc. According to Supplementary Figure 3f, the performance on Eukaryota and Fungi were good, while not quite good on Bacteria. Please explain the reason why.
12. Page 6, Line 202 - 205: How does this subsampling approach affect the accuracy? How to handle the other unselected proteins?
13. Page 6, Line 196 - 197: What does the threshold of 0.2 mean? Is that a parameter of the MAFFT software? (please clarify)
14. Page 6, Line 212-214: Does the “merging decisions” refer to the subsequent paragraph about HOG merging? Second, if the child subtree is excluded, then how will it be handled in the subsequent steps? Does it mean the child subtree (and the corresponding leaves) will not be included in any resulting orthology group?

Remarks on code availability:

I have successfully installed the FastOMA software and run it on the test data set. The resulting html page looks good to me, with figures showing the statistics of the data sets and the resulting ortholog groups. The authors could improve the README on the GitHub repository:

1. The section “How to install FastOMA” mentions four ways to install FastOMA. But I had difficulty in installing because it doesn’t explain the pre-requisites clearly, including how to install nextflow, conda, docker, etc. Please add explicit command to install conda for the first way, and to install docker for the second way. Please also add the detailed descriptions for different platforms, at least Linux and Mac.
2. Confusion in the second way to install the FastOMA. The listed step also runs the analysis on the testing data set. That means if the user follows this second way to install the FastOMA, the user does not need to follow the section “How to run FastOMA on the test data”.
3. In the section of “running FastOMA on the test data”, the command provided not applicable for the user who install FastOMA by using docker or conda. Please make a note that `–profile standard` option should be properly changed for different ways of installation, e.g., to `–profile docker` if FastOMA was installed using the 2nd way and to `–profile conda` for the first way.

Version 1:

Decision Letter:

Our ref: NMETH-BC55017A

24th Jul 2024

Dear Dr. Dessimoz,

Thank you for submitting your revised manuscript “Orthology inference at scale with FastOMA” (NMETH-BC55017A). It has now been seen by the original referees and their comments are below. The reviewers find that the paper has improved in revision, and therefore we’ll be happy in principle to publish it in Nature Methods, pending minor revisions to satisfy the referees’ final requests and to comply with our editorial and formatting guidelines.

TRANSPARENT PEER REVIEW

Nature Methods offers a transparent peer review option for new original research manuscripts submitted from 17th February 2021. We encourage increased transparency in peer review by publishing the reviewer comments, author rebuttal letters and editorial decision letters if the authors agree. Such peer review material is made available as a supplementary peer review file. **Please state in the cover letter ‘I wish to participate in transparent peer review’ if you want to opt in, or ‘I do not wish to participate in transparent peer review’ if you don’t.** Failure to state your preference will result in delays in accepting your manuscript for publication.

Please note: we allow redactions to authors’ rebuttal and reviewer comments in the interest of confidentiality. If you are concerned about the release of confidential data, please let us know specifically what information you would like to have removed. Please note that we cannot incorporate redactions for any other reasons. Reviewer names will be published in the peer review files if the reviewer signed the comments to authors, or if reviewers explicitly agree to release their name. For more information, please refer to our [FAQ](https://www.nature.com/documents/nr-transparent-peer-review.pdf) page.

ORCID

IMPORTANT: Non-corresponding authors do not have to link their ORCID IDs but are encouraged to do so. Please note that it will not be possible to add/modify ORCID IDs at proof. Thus, please let your co-authors know that if they wish to have their ORCID added to the paper they must follow the procedure described in the following link prior to acceptance:

Sincerely,

Lin Tang, PhD
Senior Editor
Nature Methods

Reviewer #1 (Remarks to the Author):

The authors have done a great job in addressing my questions and concerns. FastOMA is a powerful method and hence I enthusiastically recommend publishing it.

Reviewer #2 (Remarks to the Author):

All my comments have been addressed satisfactorily. Congrats to the nice work.

Reviewer #3 (Remarks to the Author):

Overall, the authors have addressed our concerns and requests well. I believe the additional information has strengthened the paper. Moreover, the revised documentation on GitHub is now clearer.

There are a few more comments regarding the normalized Robinson-Foulds distance:

1. Lines 70-71: "...normalized Robinson-Foulds distance - the number of edges in common between two trees normalized by the total number of edges." is incorrect. Change "in common" to "different" and "total number of edges" to "total number of internal edges."
2. Lines 516-519: The description of the normalized Robinson-Foulds distance is not clear. Unsure why you need this, as it was already explained in L 70-71.
3. For some figures, such as 2b, 3a-c, 4a-d, 6a-c, 7a-d, 9a-c, and 10a-d, change the y-axis to include "normalized" when mentioning Robinson-Foulds.

Reviewer #3 (Remarks on code availability):

All good.

Version 2:

Decision Letter:

29th Oct 2024

Dear Professor Dessimoz,

I am pleased to inform you that your Brief Communication, "Orthology inference at scale with FastOMA", has now been accepted for publication in Nature Methods. The received and accepted dates will be 16th Jan 2024 and 29th Oct 2024. This note is intended to let you know what to expect from us over the next month or so, and to let you know where to address any further questions.

[REDACTED]

Over the next few weeks, your paper will be copyedited to ensure that it conforms to Nature Methods style. Once your paper is typeset, you will receive an email with a link to choose the appropriate publishing options for your paper and our Author Services team will be in touch regarding any additional information that may be required.

If you have any questions about our publishing options, costs, Open Access requirements, or our legal forms, please contact

ASJournals@springernature.com

Once proofs are generated, they will be sent to you electronically and you will be asked to send a corrected version within 48 hours. It is extremely important that you let us know now whether you will be difficult to contact over the next month. If this is the case, we ask that you send us the contact information (email, phone and fax) of someone who will be able to check the proofs and deal with any last-minute problems.

If, when you receive your proof, you cannot meet the deadline, please inform us at rjsproduction@springernature.com immediately.

Please note that *Nature Methods* is a Transformative Journal (TJ). Authors may publish their research with us through the traditional subscription access route or make their paper immediately open access through payment of an article-processing charge (APC). Authors will not be required to make a final decision about access to their article until it has been accepted. [Find out more about Transformative Journals](https://www.springernature.com/gp/open-research/transformative-journals)

To assist our authors in disseminating their research to the broader community, our SharedIt initiative provides you with a unique shareable link that will allow anyone (with or without a subscription) to read the published article. Recipients of the link with a subscription will also be able to download and print the PDF. As soon as your article is published, you will receive an automated email with your shareable link.

Please note that you and your coauthors may order reprints and single copies of the issue containing your article through Springer Nature Limited's reprint website, which is located at <http://www.nature.com/reprints/author-reprints.html>. If there are any questions about reprints please send an email to author-reprints@nature.com and someone will assist you.

Please feel free to contact me if you have questions about any of these points. Thank you very much for publishing your paper at *Nature Methods*!

Best regards,

Lin Tang, PhD
Senior Editor
Nature Methods

** Visit the Springer Nature Editorial and Publishing website at [www.springernature.com/editorial-and-publishing-jobs](http://www.springernature.com/editorial-and-publishing-jobs?utm_source=ejP_NMeth_email&utm_medium=ejP_NMeth_email&utm_campaign=ejp_Nmeth) for more information about our career opportunities. If you have any questions please click [here](mailto:editorial.publishing.jobs@springernature.com).**

Open Access This Peer Review File is licensed under a Creative Commons Attribution 4.0 International License, which permits use, sharing, adaptation, distribution and reproduction in any medium or format, as long as you give appropriate credit to the original author(s) and the source, provide a link to the Creative Commons license, and indicate if changes were made. In cases where reviewers are anonymous, credit should be given to 'Anonymous Referee' and the source. The images or other third party material in this Peer Review File are included in the article's Creative Commons license, unless indicated otherwise in a credit line to the material. If material is not included in the article's Creative Commons license and your intended use is not permitted by statutory regulation or exceeds the permitted use, you will need to obtain permission directly from the copyright holder.

Response to Reviewers' Comments:

Reviewer #1:

Remarks to the Author: The manuscript presents FastOMA, a new method building on the OMA framework, to infer orthologous genes across large datasets.

The most important feature is the linear scalability with respect to the number of input species, which makes FastOMA a method that can handle thousands of genomes. At the heart of this speed improvement is replacing the all-vs-all alignment step with a k-mer based grouping of new genes into existing homology groups (OMAmer) and efficient parallelization over the ortholog groups and the phylogenetic tree. The authors show that FastOMA achieves a high accuracy, comparable to the best methods available. The method can also handle isoforms by picking the one that best matches a protein in the reference HOG, and benefits from more accurate phylogenies.

The method is an asset in an era where hundreds of genomes will (hopefully) soon be generated per day. The paper is well written and well explained. I only have minor comments and suggestions.

RESPONSE: We thank the reviewer for their very positive overall assessment, and for their constructive points, which we address in detail below.

REVIEWER COMMENT: My major question is how the method works in case the input tree has a polytomy (not sure if a strictly bifurcating tree is expected; the NCBI taxonomy (line 85) is certainly not) or has an error. E.g. if the given tree includes ((A,B)C) as a clade, but in reality it is (A(B,C)), this will likely cause an error in the gene tree reconciliation step for A+B. Would this get fixed when A+B+C is considered as a clade, or would the early error be propagated up the tree. Maybe robustness to tree error could also be tested by introducing errors in the input tree, and determining how accuracy drops.

RESPONSE: We thank the reviewer for raising this important point regarding the impact of the species tree on FastOMA's orthology inference.

First of all, indeed, FastOMA does not require an input species tree that is fully resolved or strictly binary. The degree of resolution matters: we show in Figure 2d that using a more resolved species tree leads to improved orthology calls (with fewer induced gene losses).

The accuracy of the species tree does influence the estimation of orthologous and paralogous groups in FastOMA. This is because the species tree is used to determine the order in which gene comparisons are made and to define all the internal nodes (i.e. ancestral species) for which HOGs (Hierarchical Orthologous Groups) are inferred. Thus, errors in the species tree can potentially propagate through the orthology inference process. For example, if a duplication occurred in species A and B but not in C, and the tree incorrectly places B and C as closer relatives, this can lead to incorrect inferences about duplication events and orthologous relationships. With a wrong species tree, when B and C are compared first, a duplication is inferred only inside B, with the single gene of C as an outgroup. This creates a single subHOG at the (wrong) taxonomic level B+C. When A is added, the subHOG will become split between the two copies found in A, and we would not recognize the correct orthologous relationship at the A+B+C taxonomic level.

That said, the resulting orthology/paralogy calls are quite robust, as supported by conceptual and empirical observations:

- 1) To infer the duplication nodes in gene trees, a critical step in FastOMA's orthology/paralogy delineation, we use "species overlap" and not reconciliation. Thus, instead of relying on knowledge of

the evolutionary relationships among species, this approach only relies on multiple gene copies found in the same species to infer duplication nodes (for more extensive explanations, see Gabaldon, *Genome Biology* 2008, doi:10.1186/gb-2008-9-10-235).

- 2) Although the NCBI taxonomy is imperfect, FastOMA’s good performance in benchmarks indicates that the approach is robust.
- 3) New analyses: to gain additional insights into FastOMA’s robustness to errors in the species tree, we have conducted a supplemental experiment where we introduced dramatic errors in the species tree—swapping branches—and investigated how these errors impact the accuracy of FastOMA’s predictions using the QfO benchmark. We randomly selected five pairs of species (i.e. 10 out of the 78 species) and swapped each species pair in the species tree. The normalised Robinson-Foulds (RF) distance between the resulting trees and the true QfO species tree were 0.49 and 0.33 for “SwappedTree1” and “SwappedTree2”, respectively. This means that nearly half of the branches in SwappedTree1 were spurious.

Here, we provide the results for the benchmarking of species discordance tests for LUCA/Vertebrata, Agreement with reference gene phylogenies. The rest of the benchmarks are provided in Supplementary Figures 18-20.

Overall, this major perturbation of the input species tree did result in some performance deterioration, but to a relatively limited extent, with the tree with more disruption (SwappedTree1) resulting in a worse outcome.

To clarify the impact of species tree in the main text, we added the following to the manuscript: Main text, Method section: and **Supplementary Information S7**:

In the Main text:

FastOMA is also robust to errors artificially introduced in the species taxonomy (Supplementary Figure 18-20).

In the Methods section:

We also conducted another analysis to study the impact of the species tree for the QfO dataset where five pairs of species are swapped. The results are provided in Supplementary Figures 18-20, where FastOMA shows a moderate level of robustness. However, having an erroneous species tree impacted the orthology inference by introducing false positives.

In Supplementary Information:

S7. Impact of species tree on FastOMA

Since one of the FastOMA's inputs is the species tree, we investigated its impact on the orthology inference. As described in the Methods section, we swapped five pairs of species (10 species in total) in the QfO dataset. The results of benchmarking tests are shown in Supplementary Figures 18-20. Notably, the results for clade-specific species discordance tests were affected more than that of LUCA. The erroneous placement of species results as an outgroup in the gene tree, resulting in an inferred speciation event. Such an event led to merging the subHOGs, creating a bigger HOG at a level. Such a bigger HOG generates much more orthologous pairs. Overall, we could conclude that having an erroneous species tree impacted the orthology inference by introducing false positives, and FastOMA shows a moderate level of robustness.

In the Methods section:

Furthermore, FastOMA has a mechanism to handle spuriously merged subHOGs; at the deeper taxonomy level, when genes within a subHOG coalesce at a duplication event in the gene tree, FastOMA splits the subHOG into two, ensuring copies of ancestral genes are not co-present in a subHOG.

REVIEWER COMMENT: OMAMer requires reference HOGs as input and the manuscript in line 41 (current knowledge of the sequence universe) also mentions this. It would be great to make clear in Figure 1, which input are new input proteomes and which are reference HOGs (gene families).

RESPONSE: Thank you for the suggestion. We have updated the top part of Figure 1 as below to clearly distinguish between new input proteomes and reference HOGs (gene families):

We also clarified in the Methods section that the proteins in the input reference HOGs are only used in the initial step of family delineation, which places the input proteins into rootHOGs. After this initial step, the

proteins in the database reference HOGs are not used in the subsequent steps in FastOMA. In other words, only the query proteins from the input FASTA files are used in HOG inference.

Proteins mapped to the same reference HOG are then grouped together, forming query rootHOGs, with the exclusion of proteins already present in the database. Thus, proteins in the database reference HOGs are not used in the next steps in FastOMA.

REVIEWER COMMENT: line 63. It would be more informative to report the number of cores (not CPUs), as CPUs nowadays have many cores. In line 297 the authors likely mean 96 nodes each equipped with an AMD CPU with 48 cores.

RESPONSE: We have updated the text as follows to clarify the number of CPU cores:

All the analyses were conducted on the high-performance computer cluster of the University of Lausanne which houses 96 computation nodes. Each node is equipped with two 24-core AMD CPUs, totaling 48 cores per node.

FastOMA inferred orthology among all 2,086 eukaryotic UniProt reference proteomes in under 24 hours, using 300 CPU cores.

REVIEWER COMMENT: Line 28: Scalability issues and all-vs-all. While this is true for almost all methods, there is one orthology inference method available that scales linear with the number of species (<https://www.science.org/doi/full/10.1126/science.abn3107>)

RESPONSE: Thanks for pointing this out. Indeed, if whole genome alignments are available, TOGA is indeed a compelling and efficient approach. We now mention TOGA in the introduction of the manuscript:

Methods relying on whole genome alignment, such as TOGA⁷, are more efficient but the genome alignment requirement limits their applicability to relatively closely related species.

Kirilenko, Bogdan M., et al. "Integrating gene annotation with orthology inference at scale." *Science* 380.6643 (2023).

REVIEWER COMMENT: Line 626 mentions that fragmented gene models can arise due to genome annotation errors. Another likely source is assembly fragmentation, where the gene is split across 2 or more scaffolds. While this rarely a problem for long-read based assemblies, most of the existing assemblies are based on short reads and this error is quite prevalent.

RESPONSE: That is a good point. We included this point in the manuscript, section S8.

This is done to correct issues that might arise due to errors in genome annotation or to fragmented genome assembly where a gene is split across separate scaffolds.

REVIEWER COMMENT: As a note, methods like FastOMA will benefit from more comprehensive

genomes covering different taxonomic groups, as this will make k-mer approaches more efficient (simply by reducing the sequence divergence between species pairs) and reducing the number of proteins not assigned to a reference HOG. This forward-looking point could be mentioned.

Along a similar line, phylogenetic resolution and thus input tree knowledge will improve when more species have been sequenced.

RESPONSE: We share the reviewer's positive outlook on how FastOMA will benefit from a higher coverage database. In the OMA team, we provide regular releases to the OMA browser (at least once a year), with which OMAMer works. As our database grows, OMAMer yields higher-quality gene family delineations, which potentially improves FastOMA's recall.

We highlighted this point in the main text as below:

FastOMA can thus use advances in taxonomic knowledge for better orthology predictions and will benefit from the higher resolution that is brought by new genomic sequences from large-scale sequencing projects.

Notably, the OMA team provides regular updates to the OMA database, increasing the number and diversity of species included in the database used by OMAMer. This results in higher resolution for k-mer based grouping. As more taxa get included, we foresee FastOMA's inference will improve as more sequences are placed into rootHOGs.

REVIEWER COMMENT: Remarks on code availability:

While I didn't test FastOMA myself, the github is very well documented and provides test data. The code looks reasonably well documented and clean. Various options are available for running it via docker, singularity or on clusters via Slurm.

RESPONSE: Thanks for the positive feedback. We are glad that the reviewer found our tool well-documented and we are happy to provide a variety of options for running FastOMA to suit our users' needs.

Reviewer #2:

REVIEWER COMMENT: The manuscript describes the FastOMA method for orthology inference. Input is a set of proteomes, each comprising the protein sequences encoded in the genome for a certain species, and a species tree representing the evolutionary relationships between the input species. The output is a hierarchy of clusters of orthologous proteins, with each hierarchy level corresponding to an ancestral node in the species tree.

Crucially, FastOMA is the first software tool capable of inferring orthology groups for thousands of proteomes, thanks to the fact that its computation time scales only linearly, instead of quadratically as the state-of-the-art tools, with the number of input proteomes (Fig. 2c). The key idea is to make use an existing database of orthologous groups to which each sequence is mapped, avoiding all-versus-all searches. Only sequences that cannot assigned to an existing orthology group are clustered to found new clusters of orthologs.

FastOMA's unequalled speed and scaling will be critical to analyze the hundreds of thousands of eukaryotic species that are going to be sequenced over the next years, for instance within the Earth Biogenome project.

The tool is very well described in the online methods and supplemental methods sections.

The recall and precision of the inferred orthology relationships are benchmarked using the Quest for Orthologs suite of benchmark by running the publicly available benchmarking scripts with the 2020 version of QfO proteomes. While this code was developed by the same lab of Dessimoz (Altenhoff, A. M. et al. Standardized benchmarking in the quest for orthologs. *Nat. Methods* 13, 425–430 (2016)), the benchmarks have been developed jointly by the Quest for Orthologs community and approved by them. Therefore, I trust that the benchmarks afford a nuanced and balanced assessment of the various orthology inference tools.

The performance of FastOMA in the various QfO benchmarks is shown in the main text and many plots in the supplemental material. Generally speaking, FastOMA is comparable in performance to OMA and is at or near the Pareto frontier of a large number of tools benchmarked by QfO.

In summary, FastOMA is the first tool that scales linearly in the number of input proteomes. It is the only tools that can process thousands and probably even tens of thousands of proteomes while maintaining a competitive sensitivity-recall performance. Its speed leaves no alternatives when huge datasets of proteomes are to be analyzed at once.

RESPONSE: We thank the reviewer for their very positive overall evaluation and the detailed comments on our manuscript.

REVIEWER COMMENT: Minor points:

1. How much does FastOMA's performance depend on the existing OMA database? It would be insightful to run FastOMA without this database on a few benchmark datasets to assess the contribution of the OMA database to its accuracy and how much it depends on the overlap of the input proteome set with the proteomes in the OMA database.

RESPONSE: Thank you for this question. In principle, it is possible to run FastOMA entirely without relying on the OMA database, either by using reference HOGs from a different source in the family delineation step, or even by using an alternative family delineation method.

To illustrate this point and address your question about the impact on performance, we undertook additional analyses using the InterProScan package to group the QfO proteomes' genes into predefined InterProScan gene families. We then ran the rest of the FastOMA pipeline on these groups. The benchmarking results are provided in Supplementary Figures 15-17. InterProScan-based results tended to have lower specificity but

higher recall, with points often falling a bit further from the *Pareto* frontier (the line indicating the set of best performing methods along the precision-recall trade-off) than the OMA-based delineation.

Importantly, the results demonstrate that FastOMA can produce useful results even without relying on the OMA database.

We added the following text to the Methods section and Supplementary Information S6:

Analysis of the QfO reference proteome set using InterProScan classification of protein families

To study the influence of OMA database and OMamer on the performance of FastOMA, we replaced the first part of the procedure, normally done by placing query genes into the OMA database rootHOGs with OMamer, with InterProScan. We used InterProScan to group the QfO proteomes into gene families predefined by InterProScan. To do so, we first ran InterProScan with the argument *-appl Pfam* on the QfO dataset which grouped the proteins into InterProScan families. Then, we created the rootHOG with those groups, maintaining the same InterProScan family identifier. Then, we ran the rest of FastOMA on these rootHOG FASTA files. The QfO benchmarking results are shown in Supplementary Information S6 and Supplementary Figures 15-17.

[Supplementary Information] S6. Impact of OMA database on FastOMA

In this section, we describe the analysis to investigate how FastOMA is impacted by the reference OMA database. This is implemented by replacing the first step of FastOMA (*i.e.*, OMamer, which uses the OMA database) with the InterProScan package. The protein grouping (*i.e.*, finding gene families) is a crucial step because if two orthologous genes are mapped to two different groups, there is no chance of rescuing these pairs in subsequent FastOMA steps. This results in false negatives, lowering the recall in orthology inference. Note that FastOMA's grouping is a crucial step for achieving the speed; FastOMA only compares proteins that are inside the gene family, in contrast to other methods that do all-vs-all comparisons.

The benchmarking results provided in Supplementary Figures 15-17 show higher RF distance values and fraction of incorrect trees in most of the discordance tests when comparing FastOMA (OMA database) and FastOMA (InterProScan). However, FastOMA (InterProScan) was able to report more orthologous pairs and better recall. In reference gene families benchmarks, such higher recall was achieved at the expense of a drop in positive predictive values. Over all benchmarks, this strategy leads to overall higher recall but lower accuracy.

REVIEWER COMMENT: 2. As FastOMA's performance will depend to some extent on the size and completeness of OMA, it would be helpful to discuss the procedure and frequency of updates of the OMA database.

RESPONSE: The OMA browser is updated regularly, with several species added in each release. These updates are based on user feedback collected via an online form available on the website (<https://omabrowser.org/oma/suggestion/genome/>) and with a focus on underrepresented taxa (<https://omabrowser.blogspot.com/>). As an example, the July 2023 release focused on prokaryotes, with 261 prokaryotic and 19 archaea species added. We have added the following to the manuscript addressing the comment.

Notably, the OMA team provides regular updates to the OMA database, increasing the number and diversity of species included in the database used by OMAMer. This results in higher resolution for k-mer based grouping. As more taxa get included, we foresee FastOMA's inference will improve as more sequences are placed into rootHOGs.

REVIEWER COMMENT: 3. Misspelled Schilcker => Schlicker; evolutionar*il*y conserved; Fig. 1a and 1b referred to but a and b labels missing in figure 1.

RESPONSE: Thanks. We have corrected these points.

Reviewer #3:

REVIEWER COMMENT: This manuscript presents FastOMA, a reimplementation of OMA with a focus on speed while maintaining the accuracy of OMA. This is obtained by a new algorithm to reduce the time complexity from quadratic to linear in terms of the number of sequences. However, I have several concerns/comments below.

RESPONSE: We appreciate the constructive comments and feedback from the reviewer. We revised the manuscript according to these inputs and hope that we have addressed all the concerns.

REVIEWER COMMENT: 1. Given that OMA is not as popular as other methods like OrthoFinder, I'm wondering what would be the impact of FastOMA? For example, apart from speed, what other advantage(s) do you have in FastOMA compared with OrthoFinder and BUSCO? While the speedup is impressive, I believe the authors should provide other arguments to justify its publication in Nature Methods.

RESPONSE: Thank you for this very relevant question. In general, it is true that speed alone may not always be an overriding consideration. However, there are several reasons why the speed breakthrough achieved by FastOMA is so compelling:

- 1) The issue of scalability has been specifically highlighted as one of the main challenges of the field by the broad *Quest for Orthologs* community for many years (e.g. doi.org/10.1093/bioinformatics/btu492, doi.org/10.1093/molbev/msab098).
- 2) FastOMA achieves its speed up without *sacrificing the high accuracy* of the classical OMA method. Arguably, the reason OrthoFinder is currently more popular than OMA is precisely because it is so much faster. Just consider the enormous difference in Figure 2c: for many datasets, it is simply not feasible to use the classical OMA algorithm (“OMA GETHOGs”). Even so, OMA continues to be used by researchers who value high precision and rich functionality.

- 3) FastOMA achieves not only higher speed, but higher scalability. Consider again the figure above—FastOMA scales linearly, whereas the existing methods, including OrthoFinder2, scale quadratically and thus hit the wall at a few hundreds of genomes. The magnitude of this “leapfrog” is quite unusual. See also Reviewer #1’s general comments.
- 4) The unprecedented speed and scalability of FastOMA is not merely a convenience: it opens up the possibility to analyse more data. This capability significantly expands the scope of comparative genomics, allowing researchers to study evolution at the Tree of Life scale. Such large-scale analyses

were previously impractical due the limitations of existing tools. See also Reviewer #2's general comments.

- 5) FastOMA also offers additional compelling features which are not present in methods such as OrthoFinder, including the handling of isoforms and gene fragments, and detailed outputs such as gene trees and multiple sequence alignments. As part of the revision, we have added the last one (reporting gene trees and multiple sequence alignments) as a new feature to the FastOMA software to improve its functionality.
- 6) We also highlight that we provide a collection of downstream analysis tools, software packages, bulk download possibilities, APIs, interactive visualisations and several other resources, which is not offered by the sole software or other databases. Users can leverage this "OMA Ecosystem" for a diverse range of applications, which can be done on an even larger scale thanks to FastOMA.

As for BUSCO, the approach is not really comparable because it inherently focuses on universal, single-copy genes.

We have revised the manuscript to further clarify these points:

Note on parallelisation

Scalability has been a major challenge in the field of orthology inference highlighted by the Quest for Orthologs community for many years. FastOMA is optimised to process taxonomic levels in parallel (when possible) by inferring HOGs at all taxonomic levels...

[FastOMA outputs]

... Besides, the user can store the gene trees and multiple sequence alignments of the subsampled HOGs for all taxonomic levels.

[Methods]

Note that a user can provide their own initial grouping of proteins to be used with FastOMA. This could be put in practice in two ways: 1) running the last two processes of *FastOMA.nf* (*hog_rest* and *collect_subhog*) on the user's protein family in FASTA format. 2) Providing group mapping of proteins in OMamer format.

REVIEWER COMMENT: 2. Figure 1 is hard to follow and the current caption doesn't really help. The authors should spend more effort to explain it, either in caption or main text or both.

RESPONSE: Thank you for this suggestion. The Online Methods section (appeared after the references) provides a detailed explanation of the FastOMA methods, where all parts of the Figure 1 are described. The *Brief Communication* manuscript format is indeed extremely compact, but we provide details for the interested readers in the extensive supplementary materials.

We have revised the caption of Figure 1 as follows:

Figure 1. FastOMA algorithm overview. Input proteomes are mapped to reference gene families using the OMamer software. HOGs are inferred using a "bottom-up" approach, starting from the leaves of the species tree and moving towards the root. At each taxonomic level, HOGs from the child level are merged, resulting in HOGs at the current level. To decide which HOGs should be merged, sequences

from the child HOGs are used to create a multiple sequence alignment, followed by gene tree inference to identify speciation and duplication events. Child HOGs are merged if their genes evolved through speciation (see Methods section & Supplementary Information for details).

REVIEWER COMMENT: 3. Figure 2 is very complicated. For example, 2a is very busy because authors try to condense two accuracy statistics into one single figure. I would suggest to either remove several methods (e.g., keeping only those in Fig 2c) or changing it to just show one statistic on the y-axis and different methods on the x-axis (such as a barplot) or both, which will show the comparison between methods much clearer. Whereas you can move the plot for all methods into the supplement.

RESPONSE: Thank you for the feedback. We understand the concern regarding the complexity of Figure 2.

Several of these orthology methods are widely used, and we believe it is important to provide a comprehensive comparison of these methods for general readership. Each tool offers a unique trade-off in terms of precision and recall, and in several other publications on orthology inference (including SonicParanoid2: doi.org/10.1101/2023.05.14.540736, Domainoid: doi.org/10.1186/s12859-019-3137-2, and OrthoFinder: doi.org/10.1186/s13059-019-1832-y), these tools are compared. As such, some readers may expect to see these comparisons. The most important part of the figure is the Pareto line (grey dashed line) and how far each method is from this line.

To make Figure 2a easier to read and less crowded, we have adjusted the colour of the error bars to be less prominent. Furthermore, we have optimised the marker choice to improve legibility. We hope that these changes make it easier to provide a fair comparison among tools.

REVIEWER COMMENT: 4. Fig 2a shows that FastOMA is not as accurate as OrthoFinder (at least in terms of true positive rate – the x-axis). I believe this should be stated in the ms.

RESPONSE: Thank you for pointing this out. There is indeed a trade-off between positive predictive value and true positive rate, and in terms of true positive rate, OrthoFinder does particularly well on this benchmark. We have amended the text to clarify this point:

... on the SwissTree reference gene phylogeny benchmark, FastOMA outperforms other methods with a precision of 0.955 in reference gene phylogenies (Figure 2a). With a recall in line with most state-of-the-art methods (0.69, lower than those of Panther and OrthoFinder), the balance of these metrics indicates a well-tuned approach to orthology inference, with a focus on minimising false positives.

REVIEWER COMMENT: 5. I don't understand fig 2b and 2d.

RESPONSE: Thank you for pointing at parts of the manuscript that require clarification. We have updated the text and the figure captions to refer the reader to the relevant part in the text. We also added references on how the benchmark in Figure 2b was defined in the Supplementary Information S2.

In a nutshell, the idea behind these benchmarking tests is to provide a proxy for comparing the performance of different tools since there is no ground truth for orthology information. Importantly, there is no single best method, as each method offers a trade off between recall and precision for each benchmarking test. However, tools that appear on the Pareto frontier are considered optimal in terms of recall/specificity trade-off.

Specifically, we now describe the Generalised Species Discorance Test provided in Figure 2b in Supplementary Information S2.1. This test estimates the amount of agreement between the gene tree from orthologous genes and a known species tree, which should be high if the genes are indeed orthologs. We also add the reference where these tests were originally presented and described in more detail.

Altenhoff, Adrian M., et al. "Standardized benchmarking in the Quest for Orthologs." *Nature Methods* 13.5 (2016): 425-430.

The subplots a-c in this figure are dedicated to the number of trees that were completed (using orthologous pairs as a proxy for recall) and for which Robinson-Foulds (RF) distance (to compare the topological differences between two trees) were calculated. We use the ETE3 package to calculate this metric. In principle, to calculate the RF distance between two trees, first, each tree is cut into two subtrees each time by removing a branch. Then, the sets of leaves in each partition are found and compared to the partitions of the partitions of the other tree. The number of different partitions is reported as RF distance. In other words, RF distance shows how many splits are present in a tree but not the other one. The value is normalised by dividing it by the total number of present splits.

Regarding Figure 2d, the plot demonstrates the effect of using two different species trees for FastOMA. It illustrates how parsimonious the gene families evolution is. Specifically, we measured parsimony by using the amount of implied gene losses for that family, which is usually used in the gene tree and species tree reconciliation analysis (*e.g.*, doi.org/10.1186/1471-2105-13-S19-S15). Our result shows that FastOMA can benefit from a more resolved tree, resulting in more parsimonious results (*i.e.*, larger gene families with fewer inferred losses). We also mentioned the Pyham tool in the Methods section (doi.org/10.1093/bioinformatics/bty994) we used for calculating this measure.

We clarified the Figure 2d caption as follows:

Figure 2. FastOMA is not only fast but also accurate: a) Quest for Orthologs (QfO) benchmark, agreement with SwissTree reference phylogeny; b) QfO benchmarking of generalised species discordance test on Eukaryota clade, where the gene tree inferred from orthologous genes are compared with the reference species tree (see Supplementary Information S2.1 for detail); c) computation time comparison of FastOMA and state-of-the-art alternatives; d) impact of species tree resolution on the complexity of gene family evolutionary scenario (proxied by the number of gene

losses over the gene family history). Each point represents a gene family (a “root HOG”), whereby the size of a gene family corresponds to the number of genes in it. (The figure is truncated to focus on the most relevant region; version with all data in Supplementary Figure 24; See Methods for implied losses calculation).

We also added references to the caption and Online methods. If any point remains unclear, we would be happy to clarify further.

REVIEWER COMMENT: 6. Page 2, Line 76: Authors should explain what is “normalised Robinson Foulds distance” for non-phylogeneticists.

RESPONSE: We add the definition to the main text as suggested:

FastOMA is amongst those with the lowest topological error, with a normalised Robinson-Foulds distance– the amount of edges in common between two trees normalised by the total number of edges– of 0.225 to the reference tree, at moderate recall.

We also mention the more detailed definition of normalised Robinson Foulds distance in Supplementary Information S2 which is cited in Page 2 in the first appearance of this metric in the text.

We use the ETE3 package to calculate this metric. In principle, to calculate the RF distance between two trees, first, each tree is cut into two subtrees each time by removing a branch. Then, the sets of leaves in each partition are found and compared to the partitions of the partitions of the other tree. The number of different partitions is reported as RF distance. The value is normalised by dividing it by the total number of present splits.

REVIEWER COMMENT: 7. y-axis label of 2b is misleading: Precision is not Robinson Foulds distance, because readers would expect higher precision to mean better, which is not the case here.

RESPONSE: We thank the reviewer for pointing this out and apologize for the oversight. We corrected Precision to “Error rate” in the label of Y-axis of Figure 2b (as below).

● FastOMA	■ InParanoid_Xenfix	● OrthoMCL	▼ Hieranoid_2
▲ OMA_Pairs	➤ sonicparanoid	● OrthoInspector 3	+ OrthoFinder_MSA_v2.5.2
• OMA_Groups	✕ Domainoid+		◀ PANTHER_16_all
★ OMA_GETHOGs			▶ Ensembl_Compara

REVIEWER COMMENT: 8. Page 2, line 63: authors say it took under 24 hours using 300 CPUs, which is not clear how this is possible, because on page 8 line 296-299, they are using 96 nodes each with 48 CPUs. Is that because the authors can distribute the computation across different nodes? And how many CPU cores per node are used? Please clarify.

RESPONSE: Exactly, FastOMA is able to distribute the orthology inference task to several computation nodes (managed by SLURM scheduler), each node with 48 CPU cores thanks to the Nextflow paradigm. To clarify the number of CPUs we have mentioned the following in the text (this point was also mentioned by the other reviewer).

In main text:

FastOMA inferred orthology among all 2,086 eukaryotic UniProt reference proteomes in under 24 hours, using 300 CPU cores.

In Methods section:

All the analyses were conducted on the high-performance computer cluster of the University of Lausanne which houses 96 computation nodes. Each node is equipped with two 24-core AMD CPUs, totaling 48 cores per node.

In Supplementary Information S11:

As an example, consider a dataset with 10 species and 1004 reference gene families. First, the job *check_input* with one CPU core is submitted. Once finished, 10 *OMAmer* jobs each with one CPU core will be run. *FastOMA.nf* waits until all of them are done. Then, using one CPU core, *infer_roothog* groups the proteins into 1004 families. Next, using one CPU core, *batch_rootHOG* generates (for example) 5 batches of small rootHOGs (each of which includes 200 rootHOGs) and 4 big rootHOGs. Now, *infer_subHOG* (under the wrapper *hog_rest*) starts for the 5 batches (each with one CPU core) and 4 big rootHOGs (under the wrapper *hog_big*, each with 6 CPU cores) totaling 5+24=29 CPU cores. Once all batches are finished, *collect_suhog* reads the output of each batch and writes the final OrthoXML file.

If the total number of needed CPU cores exceeds 500, Nextflow submits the first 500 jobs and sequentially adds one more job, once a job finishes. Start of a job relies on the availability of computation nodes assigned with Slurm Workload Manager on a High Performance Computing cluster. When a job finishes with an error due to time/memory limits, FastOMA re-submits it by doubling the CPU, time or memory. FastOMA also tries to decide the number of CPU cores and memory based on the size of the gene families in the batch. The number of CPUs per task can be changed with the Nextflow configuration file available at our GitHub *FastOMA/conf/base.config*.

In short, FastOMA tries to parallelise the steps as much as possible (with the goal of optimising the wall-clock time) considering the dependencies, using a different number of CPU cores for each step, specified based on the number of species and gene families.

REVIEWER COMMENT: 9. Figure 2c: I would show the wall-clock times instead of CPU times, because it gives readers an impression that it takes 3000 hours (125 days) to get the results, but in fact only 24 hours to wait for the result as written in the main text.

RESPONSE: Thanks for the suggestion. We have added the comparison of wall-clock time as Supplementary Figure 25. We decided not to include this figure in the main text because the Diamond part of OrthoFinder and all steps of FastOMA use different nodes on the cluster. As a result, the reported wall-clock time might have been affected by the availability of nodes at the time of each run. However, the CPU times reported in Figure 2c are more accurate.

We updated the text to say:

The comparison of tools in terms of wall-clock time in hours is reported in Supplementary Figure 25. The Diamond part of OrthoFinder and all steps of FastOMA use different nodes on the cluster, so the reported wall-clock time might have been affected by the availability of nodes at the time of each run. However, the CPU times reported in Figure 2c are more accurate.

Supplementary Figure 25. The comparison of tools in terms of Wall-clock time in hour.

REVIEWER COMMENT: 10. Subfigures in Figure 2 should reordered by reference in the main text. For example, 2c should become 2a as 2c is mentioned first in the text.

RESPONSE: Thank you for the comment, which is a valid point as readers typically expect to see the figures in the order they are referenced in the text. Since subfigures 2a and 2b share the same legend, to save space and maintain a better layout, we prefer to keep the current order on the figure. However, we revised the text to match the order of the subfigures in the figure.

REVIEWER COMMENT: 11. Page 13, From the supplementary, the experiments were performed in considerable large scale, with over 984K protein among 78 species, including Eukaryota, Fungi, Bacteria, etc. According to Supplementary Figure 3f, the performance on Eukaryota and Fungi were good, while not quite good on Bacteria. Please explain the reason why.

RESPONSE: Supplementary Figure 3f reports the number of predicted orthologs (x-axis) against the fraction of incorrect trees (y-axis), which indicates the proportion of gene trees that have at least one difference from the species tree. FastOMA performs noticeably worse than the methods on the Pareto frontier on this particular benchmark. However, it is important to note that the fraction of incorrect trees is also noticeably higher in Bacteria than in the Eukaryotic clade for all methods (0.85-0.95 for Bacteria vs. 0.05-0.15 for Eukaryota). This should be compared to Supplementary Figure 3c, which reports the same results but uses average RF distance as the metric. The average RF distance provides a higher resolution comparison, with large error bars specifically for Bacteria on Supplementary Figure 3f compared to the close performance of different tools in Supplementary Figure 3c. From this, we can deduce FastOMA makes errors in more trees, but the magnitude of errors across trees is still close to that of other methods.

It is challenging to pinpoint the specific reason why FastOMA performs worse in this one particular benchmark than a few other methods, but one plausible explanation is that FastOMA assumes vertical descent of genes when inferring orthologs. Bacteria are known to be subject to Horizontal Gene Transfers, which breaks this assumption. This likely explains the overall higher incongruence of gene trees to species trees observed for every method, and particularly for FastOMA.

To address this, we have revised the relevant part in Supplementary Information S2.1 as follows:

In this benchmark (Supplementary Figure 3a-c), FastOMA performs well, with a low average Robinson-Foulds distance (higher precision) and moderate recall, as reflected in the number of completed tree samplings and number of orthologous pairs. This places FastOMA close to the Pareto frontier for three clades of Eukaryota, Fungi, and Bacteria.

The subplots d-f in Supplementary Figure 3 show the same benchmark data, but use the fraction of incorrect trees (gene trees with at least one difference from the species tree) as a measure of accuracy and the number of orthologs as a measure of recall.

FastOMA results are similarly close to the Pareto frontier with this measure for Eukaryota and Fungi. It does not perform better than other methods in Bacteria, with the proportion of incorrect trees being higher than other methods with a similar number of orthologs. This might be related to the fact that the FastOMA algorithm for orthology calling assumes vertical descent between genes, an assumption that is not always met in Bacteria.

REVIEWER COMMENT: 12. Page 6, Line 202 - 205: How does this subsampling approach affect the accuracy? How to handle the other unselected proteins?

RESPONSE: Let us first answer the second question. The unselected proteins are not included in the alignment and tree computations, but they are not lost—they are “represented” by the sampled proteins. Consider for instance the human p53 tumor suppressor protein. There are many orthologous p53 proteins in other primate species, which will be captured in the p53 HOG at the primate level. However, to establish that primate p53 proteins have diverged from p63 and p73 proteins near the common ancestor of the bony fish, it may be sufficient to keep just a few p53 primate sequences in the analysis. Thus, when inferring the nested HOG structure, the unsampled sequences will have the same fate as the rest of the proteins in the same group at the defined taxonomic level.

To assess how the subsampling process affects the results, we conducted a new analysis where we varied the subsampling values to 5, 20, 100 proteins and no subsampling on the QfO dataset. The QfO benchmarking

results are shown in Supplementary Figures 12-14. Our results show that changing the number of proteins sampled for each subHOG results in only a slight change in orthology accuracy (as an example please see below, Supplementary Figure 12a).

To clarify this for the readers of the manuscript, we have added the following to the Methods, section Step 2) FastOMA orthology inference:

...we implement a subsampling approach, retaining only a specified number of proteins per HOG (see Supplementary Figure 12-14, by default 20 proteins are randomly selected) used for the multiple sequence alignment (MSA) and tree inference. And, the unsampled sequences will have the same fate as the rest of the proteins in the same group at the defined taxonomic level.

The subsampling strategy is key to the speed of FastOMA, and expectedly there is a trade-off between accuracy and speed. Our benchmarking results indicate that FastOMA performs well with the subsampling approach, but users can adjust the degree of the subsampling in the parameter file.

REVIEWER COMMENT: 13. Page 6, Line 196 - 197: What does the threshold of 0.2 mean? Is that a parameter of the MAFFT software? (please clarify)

RESPONSE: To clarify this threshold we added the following to the text:

As part of the FastOMA python script, the MSA undergoes column-wise trimming with a default threshold of 0.2, meaning that we remove columns of the MSA that have more than 80% gap elements.

REVIEWER COMMENT: 14. Page 6, Line 212-214: Does the "merging decisions" refer to the subsequent paragraph about HOG merging?

RESPONSE: That's true. It refers to the next subsection titled "HOG merging". We updated the text as below:

When the species overlap ratio is less than 0.1 (as per default settings), indicating very low support for a duplication event, all leaves from the child subtree with the least number of proteins are excluded from merging decisions (described in the next subsection “*HOG merging*”). In other words, these proteins will stay in the corresponding HOGs as it was in the previous taxonomic level, and only the taxonomic label of the HOG is updated to the current taxonomic level (assuming no other merging happens in another part of the gene tree for this HOG). This is done to ensure that errors in gene annotation or inaccurate tree inference minimally affect the orthology inference.

HOG merging

Starting from the root of the gene tree, evidence of a speciation event (i.e., the internal node annotated as a speciation event due to no species overlap) prompts the merging of the HOGs of the leaves descending from the nodes.

Second, if the child subtree is excluded, then how will it be handled in the subsequent steps? Does it mean the child subtree (and the corresponding leaves) will not be included in any resulting orthology group?

RESPONSE: Here, exclusion of proteins means that they are treated separately, so these proteins will be reported in the output orthology result. To be precise, these excluded proteins are not used during the HOG merging process but these proteins will be retained in the HOG to which they belong. To clarify the algorithm, we updated the text as below:

... When the species overlap ratio is less than 0.1 (as per default settings), indicating very low support for a duplication event, all leaves from the child subtree with the least number of proteins are excluded from merging decisions (described in the next subsection “*HOG merging*”). In other words, these proteins will stay in the corresponding HOGs as it was in the previous taxonomic level, and only the taxonomic label of the HOG is updated to the current taxonomic level (assuming no other merging happens in another part of the gene tree for this HOG). This is done to ensure that errors in gene annotation or inaccurate tree inference minimally affect the orthology inference.

REVIEWER COMMENT: Remarks on code availability:

I have successfully installed the FastOMA software and run it on the test data set. The resulting html page looks good to me, with figures showing the statistics of the data sets and the resulting ortholog groups.

RESPONSE: We are glad that the reviewer was able to successfully install FastOMA and is satisfied with the FastOMA’s report. We appreciate the effort in testing our tool and checking the results.

REVIEWER COMMENT: The authors could improve the README on the GitHub repository:

1. The section “How to install FastOMA” mentions four ways to install FastOMA. But I had difficulty in installing because it doesn’t explain the pre-requisites clearly, including how to install nextflow, conda, docker, etc. Please add explicit command to install conda for the first way, and to install docker for the second way. Please also add the detailed descriptions for different platforms, at least Linux and Mac.

RESPONSE: Thank you for your feedback. We have added explicit instructions to install and use Conda and Docker in the GitHub README. Nextflow is now listed as a prerequisite, so we expect it to be installed automatically. Since Conda is a package manager that arranges installation based on the user's platform, e.g. Linux and Mac, no specific handling is needed. We have also mentioned the Conda installation for Mac and Linux separately.

However, we kept the first method as “Running the workflow directly” and listed Conda later because Nextflow itself can handle the installation using `-profile conda` which makes it easier to run FastOMA without requiring separate installation steps.

REVIEWER COMMENT: 2. Confusion in the second way to install the FastOMA. The listed step also runs the analysis on the testing data set. That means if the user follows this second way to install the FastOMA, the user does not need to follow the section “How to run FastOMA on the test data”.

RESPONSE: That's right. If the user uses FastOMA via docker, we would expect that they will see the results of the analysis on the test dataset, and therefore do not need to follow the section “How to run FastOMA on the test data” (please also see our answer to the next comment). Specifically, we updated the description and referred the user to the Section “How to run FastOMA on the test data”, where more details are provided. However, we have kept this section for users who may follow different parts of the documentation independently. This approach allows us to cater to the varying needs of different users. We hope that a wide range of users can benefit from our package and have a smooth experience in installing and using FastOMA.

REVIEWER COMMENT: 3. In the section of “running FastOMA on the test data”, the command provided not applicable for the user who install FastOMA by using docker or conda. Please make a note that `-profile` standard option should be properly changed for different ways of installation, e.g., to `-profile docker` if FastOMA was installed using the 2nd way and to `-profile conda` for the first way.

RESPONSE: That's a good point. We have corrected the documentation and refer the user to the appropriate section.

Reviewer #1:

Remarks to the Author:

The authors have done a great job in addressing my questions and concerns. FastOMA is a powerful method and hence I enthusiastically recommend publishing it.

RESPONSE: Thanks.

Reviewer #2:

Remarks to the Author:

All my comments have been addressed satisfactorily. Congrats to the nice work.

RESPONSE: Thanks.

Reviewer #3:

Remarks to the Author:

Overall, the authors have addressed our concerns and requests well. I believe the additional information has strengthened the paper. Moreover, the revised documentation on GitHub is now clearer. There are a few more comments regarding the normalized Robinson-Foulds distance:

1. Lines 70-71: "...normalized Robinson-Foulds distance - the number of edges in common between two trees normalized by the total number of edges." is incorrect. Change "in common" to "different" and "total number of edges" to "total number of internal edges."

RESPONSE: We have corrected as suggested.

with a normalised Robinson-Foulds distance – the number of different edges in common between two trees normalised by the total number of internal edges

2. Lines 516-519: The description of the normalized Robinson-Foulds distance is not clear. Unsure why you need this, as it was already explained in L 70-71.

RESPONSE: We have removed these lines since it was an extra description.

3. For some figures, such as 2b, 3a-c, 4a-d, 6a-c, 7a-d, 9a-c, and 10a-d, change the y-axis to include "normalized" when mentioning Robinson-Foulds.

RESPONSE: We have altered the y-axis as suggested.